# Enhancing Water-Deficient Potato Plant Identification: Assessing Realistic Performance of Attention-Based Deep Neural Networks and Hyperspectral Imaging for Agricultural Applications

**DOI:** 10.3390/plants13141918

**Published:** 2024-07-11

**Authors:** Janez Lapajne, Ana Vojnović, Andrej Vončina, Uroš Žibrat

**Affiliations:** 1Plant Protection Department, Agricultural Institute of Slovenia, Hacquetova ulica 17, 1000 Ljubljana, Slovenia; andrej.voncina@kis.si (A.V.); uros.zibrat@kis.si (U.Ž.); 2Crop Science Department, Agricultural Institute of Slovenia, Hacquetova ulica 17, 1000 Ljubljana, Slovenia; ana.vojnovic@kis.si

**Keywords:** hyperspectral imaging, deep learning, potato plant, water-deficiency, drought stress

## Abstract

Hyperspectral imaging has emerged as a pivotal technology in agricultural research, offering a powerful means to non-invasively monitor stress factors, such as drought, in crops like potato plants. In this context, the integration of attention-based deep learning models presents a promising avenue for enhancing the efficiency of stress detection, by enabling the identification of meaningful spectral channels. This study assesses the performance of deep learning models on two potato plant cultivars exposed to water-deficient conditions. It explores how various sampling strategies and biases impact the classification metrics by using a dual-sensor hyperspectral imaging systems (VNIR -Visible and Near-Infrared and SWIR—Short-Wave Infrared). Moreover, it focuses on pinpointing crucial wavelengths within the concatenated images indicative of water-deficient conditions. The proposed deep learning model yields encouraging results. In the context of binary classification, it achieved an area under the receiver operating characteristic curve (AUC-ROC—Area Under the Receiver Operating Characteristic Curve) of 0.74 (95% CI: 0.70, 0.78) and 0.64 (95% CI: 0.56, 0.69) for the KIS Krka and KIS Savinja varieties, respectively. Moreover, the corresponding F1 scores were 0.67 (95% CI: 0.64, 0.71) and 0.63 (95% CI: 0.56, 0.68). An evaluation of the performance of the datasets with deliberately introduced biases consistently demonstrated superior results in comparison to their non-biased equivalents. Notably, the ROC-AUC values exhibited significant improvements, registering a maximum increase of 10.8% for KIS Krka and 18.9% for KIS Savinja. The wavelengths of greatest significance were observed in the ranges of 475–580 nm, 660–730 nm, 940–970 nm, 1420–1510 nm, 1875–2040 nm, and 2350–2480 nm. These findings suggest that discerning between the two treatments is attainable, despite the absence of prominently manifested symptoms of drought stress in either cultivar through visual observation. The research outcomes carry significant implications for both precision agriculture and potato breeding. In precision agriculture, precise water monitoring enhances resource allocation, irrigation, yield, and loss prevention. Hyperspectral imaging holds potential to expedite drought-tolerant cultivar selection, thereby streamlining breeding for resilient potatoes adaptable to shifting climates.

## 1. Introduction

The potato holds a prominent position among the world’s most important food crops, ranking fourth in terms of cultivation area, covering 17.6 million hectares and boasting an annual production estimated at 386 million tons [1]. Its high productivity renders it suitable for cultivation across diverse landscapes, ranging from agriculturally rich regions to more challenging environments. Consequently, the potato plays a crucial role in mitigating global hunger, supporting food security, and enhancing the livelihoods of small-scale food producers [2]. This aligns with the 2030 Agenda for Sustainable Development Goals, particularly the Zero Hunger Goal [3]. Achieving these objectives is challenging, as various stress factors hinder progress. Plant stress is often caused or facilitated by climate change; e.g., the most critical factors affecting potato yields and tuber quality are heat and drought [4,5]. Drought stress elicits a complex response in plants, influencing their growth, physiology, and metabolism. Potato (*Solanum tuberosum* L.) displays distinct reactions to water scarcity, where the impact varies depending on rate of progression, i.e., a combination of duration and intensity [6]. As water becomes limited, potato plants undergo a series of adaptive changes, e.g., closing stomata to reduce water loss through transpiration [7]. This can lead to decreased photosynthesis and growth, affecting overall yield [8,9]. Additionally, drought triggers the synthesis of stress-related compounds like proline and abscisic acid, which help the plant conserve water and maintain cellular integrity [10]. Root growth may also be altered to explore deeper soil layers in search of water [11]. While these responses help potato plants endure drought conditions, prolonged stress can ultimately impair tuber development and quality. These factors collectively result in altered distribution of nutrients, slower plant growth, and reduced plant mass, ultimately leading to a significant decline in the number, size, and overall yield of tubers [12,13]. Understanding these intricate mechanisms in potato and other crops is crucial for developing drought-tolerant varieties through breeding and biotechnological approaches, ensuring agricultural resilience in water-scarce environments [14].

Various remote sensing technologies, such as LiDAR (light detection and ranging), thermal imaging, and spectral imaging [15], can help mitigate these issues [16]. Their use provides several advantages, such as reduced human effort, objective and quantitative measurements, non-destructive monitoring, early detection of plant health, cost-effectiveness, and applicability over larger areas in combination with geographic information systems (GIS). The latter also enables upscaling lab-developed detection methods to field scale [17]. Hyperspectral imaging (HSI) is one of the most frequently used techniques [18], since it was shown to produce promising results for the detection of various biotic and abiotic stressors on different crops [19,20], including potato plants and tubers [21,22]. It is a non-invasive technique that captures an object’s spectral and spatial features through imaging at various wavelengths [23]. The optimal spectral ranges for analyzing plants and vegetation are the visible range and near-infrared range (VNIR). This enables the assessment of variations in leaf pigmentation (400–700 nm) and mesophyll cell structure (700–1300 nm). To detect alterations in plant water content, longer wavelengths (1300–2500 nm) are necessary, located in the short-wave infrared (SWIR) range [18]. Hyperspectral imaging sensors capture reflected light with high spectral, spatial, and radiometric resolution, resulting in the generation of more than twenty channels per image (often exceeding a hundred). These channels are evenly spaced with a comparatively narrow bandwidth and correspond to specific predefined wavelengths [19]. Even though a large number of spectral bands typically corresponds to a greater amount of extractable information, it is crucial to acknowledge that merely having more spectral bands does not lead to improved information extraction [24]. Data quality, sensor characteristics, processing techniques used, and expertise of the analyst are all critical factors that can influence results. Additionally, the excessive number of spectral bands may give rise to Hughes phenomenon (“curse of dimensionality”), meaning it can make it difficult to identify and extract relevant information from complex, multi-dimensional data [25]. Moreover, many of these bands introduce high multi-collinearity, which may result in overfitting and subsequently reduce the overall performance of the model [26].

The emergence of deep learning has revolutionized the field of machine learning, proving to be highly effective in solving complex problems. Deep learning has outperformed traditional models in domains such as image and speech recognition and natural language processing [27]. Its superior performance has also been demonstrated in hyperspectral image (HSI) classification tasks [28,29]. One essential architecture in deep learning is the residual neural network (ResNet), which employs skip connections or shortcuts to overcome vanishing gradient problems, making neural network optimization easier [30]. ResNet’s architecture has been adopted as a backbone in several newly proposed networks for HSI classification, such as RSSAN [31], S2RGANet [32], HResNetAM [33], and A2S2K-ResNet [34]. Another crucial concept in deep learning is the attention mechanism, initially used in natural language processing and later applied in computer vision and speech processing [35]. The attention mechanism enhances important parts of the input data, while de-emphasizing the rest, allowing the network to focus more on meaningful information [25,31]. Hu et al. proposed channel-wise attention in squeeze-and-excitation network (SENets) [36], which performs dynamic channel feature recalibration. Furthermore, Woo et al. presented a [37] a convolutional block attention module (CBAM), which learns channel and spatial attention maps for adaptive feature refinement [37]. In HSI, these attention mechanisms effectively minimize the impact of collinearity between neighboring spectral bands, retaining the most discriminative bands and improving the classification model’s discriminative power and robustness [31]. Consequently, attention-based deep neural networks have achieved state-of-the art performance in various HSI classification problems [31,32,33,34,38,39,40].

While deep learning models are still relatively new and challenging for HSI classification and agriculture in general, there have been many recent applications and achievements of these models in this field. For instance, Zheng et al. [41] devised a laboratory experiment to detect drought stress in pepper leaves. Similarly, Zhang et al. [42] performed a study to classify seeds from several corn varieties and Garillos-Manliguez et al. [43] estimated papaya fruit maturity level. Meng et al. [44] designed field research to differentiate between rice varieties by using an HSI device mounted on an UAV (unmanned aerial vehicle) platform. In these studies, the datasets were acquired in a single imaging session; i.e., they didn’t consider temporal qualification of samples. In contrast, Nagasubramanian et al. [45] performed a laboratory experiment for the detection of charcoal rot on soybean plants, where the imaging sessions were performed every 3rd day after infestation was introduced. Recent studies have also delved into the detection of various biotic and abiotic stressors present on potato plants. For example, in [46], Polder et al. employed a deep neural network for virus detection and, similarly, in [47], Qi et al. employed a deep neural network with an attention module for the classification of the asymptomatic biotrophic phase of potato late blight. The datasets used in both studies were created with consequent weekly imaging acquisitions. Furthermore, in [22], Duarte-Carvajalino et al. compared several machine learning models to estimate water stress induced in potato plants. Meanwhile, in [48], Gerhards et al. investigated options for the detection of water stress expressed on plant leaves. Deep learning models for HSI classification are still underexploited in potato research [22,49,50]. Additionally, these models are applied only on hyperspectral data, which cover the VNIR part of the wavelength spectrum, whereas other parts (e.g., SWIR) could potentially provide crucial information.

In this study, we propose a new HSI-based method using attention-based deep learning for the detection of water-deficiency in potato plants. Imaging of potato plants from two varieties was performed in a time series to evaluate the classification performance over several weeks and to test the capability for early detection.

The main contributions of this paper are as follows:We propose a new deep learning network and processing pipeline for a comparatively accurate detection of potato plants exposed to water-deficient conditions.We demonstrate that both the VNIR and SWIR parts of the spectrum are relevant to effectively distinguish between well-watered and water-deficient potato plants.We implement a band attention module to extract the most important wavelengths, which provides important insights into the underlying processes.We explore various sampling schemes and show how introduced biases influence the classification performance.To the authors’ best knowledge, this is the first time that an HSI system consisting of two separate sensors, covering the VNIR and SWIR parts of the spectrum, was utilized to perform classification and selection using attention networks in agriculture.

## 2. Materials and Methods

### 2.1. Plant-Growing Setup

A greenhouse experiment (Figure 1) was set up at the Agricultural Institute of Slovenia (Ljubljana, Slovenia) from April to August 2022. Experimental conditions were carefully controlled, maintaining a temperature of 21 °C (±2 °C) during the day and 15 °C (±2 °C) at night, a relative humidity of 60% (±5%), and a photoperiod of 14 h. Tubers of potato plants (*Solanum tuberosum* L.) were planted in pots with a capacity of 5 L, such that a total of 28 and 18 potato plants of KIS Krka and KIS Savinja were planted, respectively. KIS Krka was selected as a drought-resistant and KIS Savinja as a drought-sensitive cultivar. After approximately 5 weeks of growth, half of the plants from each cultivar (14 and 9, respectively) were randomly assigned to either the drought (D) or control (C) groups (Figure 2). The plants were then cultivated for additional 5 weeks. In the drought group, the plants were subjected to a water-deficit irrigation regime. Throughout the duration of the experiment, tensiometers (14.04.04 Jett Fill tensiometers, Eijkelkamp, Giesbeek, The Netherlands) were used to monitor the moisture levels in the substrate (soil moisture). This allowed for precise control over the amount of water provided to the plants in each group. Specifically, the matric potential of the soil was carefully maintained within the range of −0.01 MPa to −0.025 MPa for well-watered plants and −0.05 MPa to −0.07 MPa for water-deficient plants. The ranges were established based on our previous experiments and expertise and were corroborated by other studies [51,52]. The exact values were checked once a day to make sure they stayed between the prescribed boundaries. Additionally, midday leaf water potential (WP) was obtained using a pressure chamber (Model 3005HGPL Soil Moisture, Inc., Santa Barbara, CA, USA) once a week at noon on eight plants, and same-day spectral imaging was performed (see next section: Section 2.2. Spectral Data Acquisition) to confirm water restriction measures.

### 2.2. Spectral Data Acquisition

Hyperspectral imaging sessions were undertaken weekly for a total of five weeks. The first imaging session was performed one week after the deficit was introduced. Hyperspectral images were acquired in the VNIR (visible to near infrared) and SWIR (short-wave infrared) spectral regions, using Hyspex (Norsk Elektro Optikk, Oslo, Norway) pushbroom cameras VNIR-1600 (400–988 nm, 160 bands, bandwidth 3.6 nm, and resolution 2700 × 1600 px) and SWIR-384 (950–2500 nm, 288 bands, bandwidth 5.4 nm, and resolution 900 × 384 px). Cameras were positioned in a dark room, 3 m above the plants. Even lighting was assured using calibrated halogen light sources, which were turned on at least 15 min prior to imaging in order to reduce light source thermal drift. Each image included up to three potato plants and a calibrated 20% reflectance panel (SphereOptics, Herrsching, Germany) on a black synthetic background with low reflectance (<5%).

### 2.3. Hyperspectral Image Preprocessing

Image preprocessing included six steps (Figure 3): (1) images were radiometrically calibrated to radiance units (W sr^−1^m^−2^) using Hyspex proprietary software (HyspexRad v3.1); (2) images were converted to reflectance using the reference panel; (3) VNIR and SWIR images were co-registered to produce one single reflectance image (as in [53]); (4) image segmentation–extraction of leaf-area pixels, whereas background pixels were set to zero; (5) extracted images were further sliced into smaller images of size 64 × 64 pixels (referred to as samples), similar to [44,45]; (6) first and last five spectral channels of both VNIR and SWIR were removed (428 bands left after removal). Reflectance values of all pixels were capped in range r, where $r\in {R:p\in \left[0,\;1\right]}^{\;}$, in case any of the pixels were saturated. With mathematical notation, the sample shape can be expressed as $h^{\;}\in R^{C\times H\times W}$, where C=428 is the number of channels and H, W=64 are the height and width, respectively. Additionally, samples covering less than 90% of the plant’s extents were removed, so number of samples varied among plants. Steps (2)–(5) were performed by using open-source software (SiaPy v0.1.1) [54,55], following methodology described in [24]. To each sample, the following attributes were assigned: “Imaging”, “Identifier”, “Treatment”, and “Variety” (Figure 3). “Imaging” denotes the week of the imaging session (1–5); “Identifier” is the unique label of the potato plant (KK-K-XX, KK-S-XX, KS-K-XX, and KS-S-XX, where XX is a number); “Treatment” defines water irrigation regime (C or D), and “Variety” is the cultivar (KIS Krka or KIS Savinja).

### 2.4. Overview of Available Data

The samples in this paper were produced using equidistant slicing of original unsegmented hyperspectral images. In total, the procedure produced 9718 samples. These were later redistributed to create the datasets, which mimic scenarios of different sampling strategies and biased sampling schemas. Of all samples, 5989 belonged to KIS Krka variety, among which 3217 and 2772 represented well-watered and water-deficient potato plants, respectively. The other 3729 belonged to KIS Savinja variety, among which 1742 and 1987 represented well-watered and water-deficient potato plants, respectively. The distribution of samples per imaging session is represented in Figure 4. Generally, the most samples were produced from imaging session 3, whereas first and last imaging sessions produced the fewest samples.

### 2.5. Preparation of Datasets and Experimental Setup

Each sample had accompanying metadata (ID, imaging session, etc.) to exactly describe to which potato plant it belongs. Based on the metadata, four datasets were prepared, separately for each variety, to investigate how data splitting and the underlying distribution affected classification outcome [56]:Unbiased dataset using stratified split (UD-SS). The number of samples was kept constant for each imaging session, and the number of samples belonging to each treatment (C or D) was equalized per imaging session. Additionally, samples from the same plant were not used both for model training and testing (i.e., a stratified split). In this setting, the model would not fit more on any of specific imaging sessions or treatments and consequently skew the results. Consequently, the effects on the results could be objectively attributed to underlying data, i.e., specific imaging session or a treatment. The same dataset was tested at the end with reduced number of spectral bands to assess how much the reduction influences the classification metrics.Unbiased dataset using random split (UD-RS). Samples were equalized, as for the unbiased dataset using stratified split. Samples for model training and testing were selected randomly, neglecting the potato plant identifier. The change in results emphasizes the importance of plant-independent splitting, i.e., if the model was overfitted on samples generated from the same potato plant.Dataset biased by treatment using stratified split (BDT-SS). The underlying distribution was skewed by different ratios between the two treatments. For KIS Krka, 20% of water-deficient samples were used from imaging 1 and imaging 2, 60% of both treatments were used from imaging 3, and 20% of well-watered samples were used from imaging 4 and imaging 5. For KIS Savinja, the selection was switched—20% of well-watered samples were used for imaging 1 and imaging 2 and 20% of water-deficient samples for imaging 4 and imaging 5. Treatment bias was introduced to show the importance of equalization of treatments through growing phase, since water-deficiency indications differ between stages.Dataset biased by imaging session using stratified split (BDI-SS). The distribution was skewed by the number of samples taken from each imaging session. For KIS Krka, 20% of samples were used from imaging 1 and imaging 2, 60% from imaging 3, and all available samples (100%) were used from imaging 4 and imaging 5. For KIS Savinja, the selection was switched; i.e., 20% of samples were used from imaging 4 and imaging 5 and 100% from imaging 1 and imaging 2. The imaging session bias was introduced to show that differentiation capability of the model may vary between imaging sessions, which translates to the overall result.

Samples from each dataset were divided into three sets: training, test, and validation. The training set contained 65% of the samples, test set 20%, and validation 15%. Number of samples for each set varied, depending on the total number of appropriate samples produced from each potato plant. The validation set was used during training for model performance evaluation, and the test set was used after training to evaluate the generalized predictive ability of the best performing model. Classification metrics were reported only for the test set. Model performance was evaluated using area under the receiver operating characteristic curve (AUC-ROC), F1 score, precision, and recall. The latter three were calculated for each treatment and averaged by support (the number of ground truth labels). The 95% confidence intervals (CIs) were calculated for all metrics, by bootstrapping with a thousand resamples, following [56].

To show how accurately the model classifies the entire potato plant, combined classification metrics were calculated using majority vote [57]. The potato plant was classified as well watered or water-deficient based on the majority of predictions that were made on initial samples covering the same potato plant; i.e., if the most samples were classified as well watered, the potato plant was classified as well watered. The metrics were calculated on pooled imaging sessions, since there were few potato plants available for testing, specifically, 15 and 10 for KIS Krka and KIS Savinja, respectively.

### 2.6. Model Architecture and Training

We implemented a convolutional neural network (CNN) with attention mechanism for identification of water-deficient plants, as shown in Figure 5. The model was constructed of five parts: (1) the network took a sample $h^{\;}\in R^{428\times 50\times 50}$ (see resizing process under augmentation techniques); (2) the sample was processed through band attention layers (see Section 2.7, Band Attention), where significance values of each spectral band were extracted; (3) by multiplying the original input sample with these significance values, a rescaled sample was produced; (4) this was fed into a residual network, which converted it through multiple residual blocks into complex latent features, and based on these outputs, a predicted value was generated; (5) the sigmoid activation function was used to constrain the calculated predicted value to range [0, 1]. Therefore, the output of the model was a continuous score, i.e., the probability of the plant being exposed to water-deficient conditions, which can then be categorized into a binary score (0: well-watered and 1: water-deficient) with a threshold value of 0.5.

The network was trained on a train set by using Adam optimization with the learning rate set to 1 × 10^−3^. It was optimized based on a binary cross entropy (BCE) loss function in a mini batch setting, where batch size was fixed at 32. During training, the performance was evaluated on a validation set. The training process lasted until the validation loss did not improve for 50 consecutive epochs (early stopping) and for maximum of 200 epochs. The trained model that achieved the lowest loss on the validation set was used on a test set to validate the model’s generalization ability. Two augmentation techniques were employed to improve the robustness and generalization ability of the model: random flips and crops [58]. In the latter, the initial spatial size of samples (64 × 64) was resized to the size expected by the network (50 × 50) to introduce stochasticity into the data and prevent the neural network from learning on noise. Using mathematical notation ${T:}^{\;}R^{428\times 64\times 64}\to R^{428\times 50\times 50}$, where T represents cropping transformation. Furthermore, learning rate decay (every 15 epochs with multiplicative factor set to 0.1) and L2 regularization (weight decay set to 5 × 10^−4^) were also utilized to compensate for overfitting. We want to stress that we evaluated several alternatives in the sense of augmentation and regularization techniques, hyperparameters for optimization (e.g., learning rate), and parameters of component building-devised deep neural network (e.g., size of convolution kernels). We used the configurations that produced optimal results as the others translated to worse or comparable results in the sense of classification performance.

In order to investigate and evaluate the performance of the model, the experiments were implemented on a computer with an Intel Xeon W-2255 CPU @ 3.70 GHz (20 cores) with 128 GB RAM and a NVIDIA GeForce GTX 3080 GPU with 12 GB of VRAM. The OS used was Windows 10 with WSL (Windows subsystem for Linux). The code was implemented in Python 3.9 programming language with additional packages. The deep learning model and optimization and evaluation architecture was implemented by using open-source PyTorch library.

### 2.7. Band Attention Mechanism

We used channel attention module [37] to adaptively recalibrate hyperspectral band channels, as proposed by Zhu et al. [31]. The module assigns higher weights to the most informative spectral bands that contain the most distinctive information, while assigning smaller weights to irrelevant or noisy bands. Therefore, band attention emphasizes spectral bands that help with feature representation and final classification. The equation summarizing attention computation can be expressed as
$$h^{l}=m(h)⊗h$$
where rescaled sample $h^{l}\in R^{C\times H\times W}$ is generated by element-wise multiplication (⊗) of spectral attention map $m\in \;R^{C\times 1\times 1}$ and input sample $h\in R^{C\times H\times W}$. Notations C, H, and W represent channel, height, and width, respectively (in our case, C = 428, W = 50, and H = 50).

To generate a spectral attention map, the spectral-wise statistics were calculated. The spectral-wise average $z_{\;}^{avg}\in R^{C\times 1\times 1}$ was generated by using global average pooling (G. avg. pool) across spatial dimensions H×W. For cth channel, it can be written as
$$z_{c}^{avg}=\frac{1}{H\cdot W}\sum_{i=1}^{H}\sum_{j=1}^{W}h_{c}(i,j)$$
where $h_{c}(i,j)$ is the value at position (i,j) of the cth channel of h. In addition, the spectral-wise maximum $z_{\;}^{max}\in R^{C\times 1\times 1}$ was also calculated, since it can provide complementary information to the average [37]. It was generated by using global maximum pooling (G. max. pool) across spatial dimensions H×W. For cth channel, it can be written as
$$z_{c}^{max}=max(h_{c}(i,j))\;\forall i,j\in Z:\;0<i,\;j\leq H,W$$

The two above-mentioned statistics ($z_{\;}^{avg}$ in $z_{\;}^{max}$) were processed with a shared network, consisting of two fully connected layers (FC), with a hidden layer set as a bottleneck. The first FC layer reduced the initial dimension by a factor of two (from vector of size 428 to 214), and the second FC layer expanded it back to initial size (vector of size 428). The transformations of both can be written as
$$s^{max}=\sigma (w_{2}(\delta (w_{1}(z^{max}))))\;s^{avg}=\sigma (w_{2}(\delta (w_{1}(z^{avg}))))$$
where the first and second FC layers are parametrized by $w_{1}$ and $w_{2}$, respectively. After each FC layer, an activation function is applied to introduce non-linearity, annotated with *δ* and *σ*, which refer to ReLU (rectified linear unit) and sigmoid function, respectively. The two outputs were then added together using simple summation over spectral dimension to generate spectral attention map:$$m=s^{max}+s^{avg}$$

The spectral attention map m was then used to transform input samples into rescaled samples.

### 2.8. Residual Network

We used the ResNet18 residual network [30] with 18 deep layers (Figure 5). The design was adopted from [59] and can be described as
$$p(h)=w_{RN}(h^{l})$$
where p∈R: p ∈ (0, 1) describes the probability of input sample h belonging to a plant exposed to water-deficient conditions. The annotation $w_{RN}$ represents the entire ResNet transformation pipeline of operations, converting the rescaled sample $h^{l}$ to a probability score. ResNet is composed of several residual blocks (RBs), where each can be described by the equation:$$h^{k+1}={f(h}^{k})+h^{k}$$
where $h^{k}$ and $h^{k+1}$ represent the outputs of the *k*th and *k* + 1th layers, respectively. Notation f represents a sequence of operations, in our case, convolutions (Conv), batch normalization (BN), and rectified linear unit activations (ReLUs). Conv layers were performed by using 3 × 3 kernels, except the first one, which used 7 × 7 kernel. After each convolution layer, BN was applied, followed by ReLU. After the first Conv layer, maximum pooling (Max. pool) with kernel size of 3 × 3 was performed to reduce initial spatial dimension. Global adaptive average pooling (G. avg. pool) was applied after the last RB, to extract an average from each channel. The calculated values were flattened to a 1-dimensional vector and fed to a fully connected (FC) layer with sigmoid activation. For additional implementation details, refer to [59] or see the provided source code (code and data availability).

### 2.9. Procedure for Selection of Prominent Spectral Bands

The attention-based model enabled learning of the weights, for band rescaling, in order to achieve better modeling performance. The spectral attention layers were used after training for effective band selection with a similar approach to the one described by Cai et al. [25]. The significance of each spectral channel was determined by averaging the band weights for all *N* samples from the training set. The average significance $r_{c}$ of the cth band can be computed by the equation:$$r_{c}=\frac{1}{N}\sum_{k=1}^{N}s_{c,k}^{avg}$$
where $s_{\;}^{avg}$ represents processed (by using the trained shared network) spectral-wise average $z_{\;}^{avg}$, created from the input sample h. Consequently, $p\in R:r_{c}\;\in \;(0,\;1),$ since the shared network constrains the output with the sigmoid function σ. Bands with larger significance values are considered to provide more information to the final classification output. In practice, the top *b* bands can be selected as a significant band subset, such that b≤C, where C is the total number of spectral channels.

It was already shown by Zheng et al. [41] that a model trained on selected spectral bands produced by attention network performs better or similarly to models trained on bands selected by any other selection techniques. That is why, in this paper, the model was retrained on as many selected bands to still achieve performance similar to without selection.

## 3. Results

### 3.1. Spectral Signatures

Average spectral signatures calculated from entire potato plants (Figure 6) exhibit high reflectance in the blue (around 400–500 nm) and red (around 600–700 nm) ranges of the VNIR part of the spectrum. In near-infrared (around 700–1300 nm), the plants exhibit a strong peak in reflectance. There are also two relative peaks in the SWIR part of the spectrum (around 1600–1900 nm and 2200–2350 nm). Noticeable differences between the two treatments are not exhibited for either variety. The ribbon representing standard deviations is generally wider for KIS Savinja. For instance, at 650 nm, the ribbon stretches for 0.12, whereas for KIS Krka, it only stretches for 0.07. Nonetheless, there are notable differences in leaf water potential measurements between the two treatments (see Table A1 and Table A2 in Appendix A). The *p*-values in these tables indicate significant differences as determined by a *t*-test.

### 3.2. Performance Evaluated on Unbiased Dataset

The performance of the proposed attention-based deep neural network was evaluated on the UD-SS. The classification metrics are shown separately for each variety in Table 1 and Table 2. The supplementary visual representation of the ROC curves is shown in Figure 7. The best results were obtained for KIS Krka, with an overall AUC-ROC of 0.74, while KIS Savinja yielded an AUC-ROC of 0.64. For KIS Krka, the model struggled in the first imaging session. In the remaining imaging sessions, it achieved AUC-ROCs between 0.71 (imaging 2) and 0.87 (imaging 4). The F1 scores were generally lower, ranging from 0.54 (imaging 1) to 0.78 (imaging 4), and precision and recall approximately followed the F1 scores. Additionally, there were no significant differences between precision and recall, indicating that the model did not overfit. The overall results were worse for KIS Savinja, with a maximum AUC-ROC score of 0.79 (imaging 2). The worst classification performance was achieved for imaging session five. Also, the model was able to classify the samples from the first imaging session with an AUC-ROC of 0.67. However, the confidence intervals for imaging 4 and imaging 5 indicate the presence of random prediction (metric of 0.5) on the lower boundary of confidence limits.

Majority voting increased the F1 scores on the pooled data for both varieties to 0.73. However, the confidence intervals were wider than before in both varieties. The samples used were the same as for the calculation of the metrics shown in Table 1 and Table 2. For KIS Krka, the AUC-ROC increased to 0.89. In contrast, the ROC-AUC metric for KIS Savinja decreased to 0.56 (Table 3).

### 3.3. Performance Comparison Evaluated Using Other Datasets

Three datasets (UD-RS, BDT-SS, and BDI-SS) with corresponding sampling schemas were compared to UD-SS to investigate the influence on the classification metrics. Table 4 and Table 5 present classification metrics evaluated on UD-RS. The model performed similarly to UD-SS for KIS Krka, as it achieved an AUC-ROC of 0.74. The values of other metrics were a bit lower, but with negligible difference; i.e., the F1 score, precision, and recall were around 0.65. In contrast, the AUC-ROC for KIS Savinja was much higher and increased to 0.87 (35.9% increase). Moreover, the F1 score, precision, and recall also increased to 0.77. Overall, for KIS KRKA, the accuracy metrics increased with time, from imaging 1 to imaging 5; the worst performance (AUC-ROC of 0.58) was achieved for imaging 1. For KIS Savinja, no such trend could be observed, and the performance was satisfactory for all imaging sessions. The AUC-ROC ranged from 0.83 (imaging 3) to 0.91 (imaging 5).

The performance of the model based on BDT-SS is presented in Table 6 and Table 7. For KIS Krka, it achieved an AUC-ROC of 0.87 (17.6% increase compared to UD-SS) and for KIS Savinja an AUC-ROC of 0.76 (18.9% increase). For KIS Krka, the F1 score reached values between 0.75 (imaging 1) and 0.84 (imaging 4) and precision and recall approximately followed. Similarly, for KIS Savinja, the model yielded F1 scores between 0.70 (imaging 5) and 0.82 (imaging 4), but it decreased substantially for imaging 3 to 0.44. Generally, a larger variation in mean values (imaging-wise) among the F1 score, precision, and recall were observed compared to UD-SS.

The model achieved an AUC-ROC of 0.82 (10.8% increase compared to UD-SS) for KIS Krka, and it yielded an AUC-ROC of 0.67 (4.7% increase) for KIS Savinja on BDI-SS presented in Table 8 and Table 9. For KIS Krka, the metrics were adequate for all imaging sessions (AUC-ROC between 0.80 and 0.95), except for imaging 1 (AUC-ROC of 0.34). For KIS Savinja, the AUC-ROC reached 0.82 for imaging 1, and 0.72 for imaging 2, but lower values for other imaging sessions, ranging from 0.28 (imaging 4) to 0.59 (imaging 3).

### 3.4. Selected Spectral Bands

The spectral channels were arranged by their relevance (Figure 8 and Figure 9) to assess which parts of the wavelength spectrum provided the most information. The selected spectral bands varied among the datasets used for either variety. However, some parts of the spectrum were more pronounced than others. For instance, the most prominent wavelengths in the VNIR part were in the ranges 475–580 nm, 660–730 nm, and 940–970 nm; and in the SWIR part, they were 1420–1510 nm, 1875–2040 nm, and 2350–2480 nm. For KIS Krka, the relevance scores of UD-SS were the most similar to BDT-SS and the most different compared to UD-RS. For KIS Savinja, the relevance scores of UD-SS were not similar to any other dataset. There were also discrepancies between varieties. For example, more significant areas were covered in the range 980–1050 nm for KIS Krka than for KIS Savinja. Regardless, relevant spectral bands were similar for both varieties, when compared to UD-SS, especially when compared to the VNIR part of the spectrum. There were, however, some minor differences in the entire SWIR part of the spectrum, with the most notable ones being in the ranges 1100–1500 nm, 1900–2050 nm, and 2350–2430 nm.

It was empirically determined that at least 50 spectral bands (Figure 10), covering the VNIR and SWIR parts of the wavelength spectrum, were needed to achieve negligible differences compared to UD-SS with all the spectral bands left intact. Generally, using fewer spectral bands translates to worse classification metrics, whereas more than 50 spectral bands cause the metrics to stay approximately constant. The detailed metrics are written in Table 10 and Table 11 for the case when the model was trained using 50 spectral bands. Specifically, it achieved an ROC-AUC of 0.73 and 0.64 and F1 scores of 0.65 and 0.61 for KIS Krka and KIS Savinja, respectively. Most metrics, calculated per imaging session, were also similar in values compared to the values generated on the full spectrum of UD-SS. However, some discrepancies still appeared; for example, the AUC-ROC was decreased to 0.29 in imaging 1 for KIS Krka. Also, the AUC-ROC was decreased to 0.55 (imaging 3) and 0.47 (imaging 4) and increased to 0.78 (imaging 1) and 0.72 (imaging 5) for KIS Savinja.

## 4. Discussion

This paper presents water-deficiency detection for potato plants through hyperspectral imaging (HSI) using a novel attention-based deep learning framework. The objectives encompassed an exploitation of the deep learning model, an exploration of sampling strategies and introduced biases, a utilization of a dual-sensor (VNIR and SWIR) HSI system, and an investigation of salient spectral channels.

Our results demonstrate that it is possible to combine hyperspectral imaging data with the proposed attention-based deep neural network to successfully discriminate between water-deficient potato plants. The challenge in distinguishing between both treatments for either variety is underscored by the similarity of their average spectral signatures (Figure 6). This highlights the potential difficulty of solely relying on spectral information for differentiation, as is the case in [22]. However, leveraging the power of deep neural networks, which exploit combined spectral–spatial information, allows for the discernment of differences based on the spatial distribution of spectral signatures within a sample [38]. Therefore, it can provide valuable information about the plant’s physiological and biochemical characteristics. For example, the spectral signatures’ relative peaks in the VNIR part emerge, because chlorophyll, the primary pigment responsible for photosynthesis, absorbs light most efficiently in the blue and red regions for energy conversion. Furthermore, increased reflectance in near-infrared is caused by cellular structures within plant leaves, such as cell walls and air spaces, which scatter and reflect light. Additionally, chlorophyll exhibits a weak absorption in this range, contributing to the high reflectance. In the SWIR range, the spectral signature changes mostly due to the presence of water and other biochemical compounds. Water molecules strongly absorb light in the first water absorption feature (around 1450 nm), which can be indicative of the plants’ water content. Also, the water is absorbed in the second water absorption feature (around 1900 nm), where it can provide insights into leaf structure and cellular water content [60]. The cellulose and lignin absorption feature (around 2100 nm) is associated with the absorption of cellulose and lignin, which are components of plant cell walls [61]. It can offer information about plant health, stress, and lignin content. At the right end of the spectral signature (beyond 2300 nm), the reflectance tends to be quite low due to various absorptions, including that of organic matter and minerals.

The classification evaluated on UD-SS was more reliable in the case of KIS Krka (AUC-ROC of 0.74 and F1 of 0.67) compared to KIS Savinja (AUC-ROC of 0.64 and F1 of 0.63), which might be due to insufficient water-deficit conditions imposed on this variety or the availability of a smaller dataset. The worst classification metrics for KIS Krka were achieved for imaging 1, which may be attributed to the low influence of the stress imposed on the potato plant or a lack of time needed for the stress to impact the spectral response. Interestingly, the opposite was true for KIS Savinja, where the worst performance was achieved for imaging 5. One possible explanation for this is the small size of the dataset, because the model did not have enough data variability to learn adequately. KIS Savinja is a drought-sensitive cultivar; hence, the response may theoretically be different (compared to KIS Krka). Nonetheless, this should only increase the classification results. The majority vote generally increased the performance (F1 of 0.73 for either variety), which is an expected outcome. However, the metrics were calculated on a small test dataset, meaning, the evaluation should be repeated on a larger dataset to firmly confirm the findings. As well, this can be seen from calculated confidence intervals, which for some metrics stretch for more than 0.50 (e.g., AUC-ROC for KIS Savinja). Nevertheless, for KIS Krka, the AUC-ROC increased to 0.89, which means that the model has a high ability to distinguish between the two varieties [62].

Deep learning models are known to learn complex features [63] and can therefore adapt to specific features of particular potato plants. Consequently, the artificial boost in classification performance (increased metrics) appeared when other datasets (UD-RS, BDT-SS, and BDI-SS) were compared to UD-SS. Therefore, we proved that the underlying distribution of samples plays another crucial aspect worth considering when evaluating deep learning models on hyperspectral data, as was also shown in other domains. This was proven many times in other domains, where these models were utilized [64]. Considering UD-RS, the calculated metrics stayed approximately the same for KIS Krka (AUC-ROC of 0.74 and F1 of 0.65) and increased for KIS Savinja (AUC-ROC of 0.87 and F1 of 0.77). The used deep learning model has a powerful modeling capability to adapt to noise incorporated in an image from which the sample was generated, causing an artificial boost in performance. The following proved that the results might become inflated if the potato plant identifier is neglected and not utilized adequately. In BDT-SS and BDI-SS, the arrangement of the samples differed between the two varieties, since the model performed better in later imaging sessions for KIS Krka and in initial imaging sessions for KIS Savinja. Furthermore, the datasets were biased in such a way that the performance had a higher chance to become artificially inflated. Both datasets for either variety achieved a higher performance compared to their non-biased counterparts, meaning that the underlying distribution of samples plays another crucial aspect worth considering when evaluating deep learning models for hyperspectral imaging.

The utilization of both hyperspectral cameras, covering the VNIR and SWIR parts of the spectrum, provided supplementary information for the detection of water-deficit conditions in potato plants. Plants undergoing drought reallocate resources to higher-potential leaves, impacting substance production and transport within plant tissues [12,52]. Drought induces metabolic changes, accumulating free sugars and essential amino acids for osmotic balance, along with a heightened production of defense compounds like protease inhibitors and oxidative enzymes [65]. The changes are highlighted by the relevant spectral ranges identified in our study, since these are known to be associated with specific physiological variables indicative of drought stress in numerous plant species [66,67]. Notably, bands correlated with leaf water content at 970 nm and 1480 nm proved helpful at estimating water deficiency, as shown before by Eitel et al. [68]. Additionally, in the VNIR part of the spectrum, two ranges, green and red, were strongly pinpointed, signifying alterations in chlorophyll and pigment contents (482–773 nm) and changes related to biomass (759–770 nm) [41]. The significant spectral bands in the SWIR part of the spectrum were associated with lignin or cellulose (2000 nm), proteins (2162–2173 nm) [69,70] and were connected to cellulose, starch, amylose, and proteins (2325–2417 nm) [71,72].

The findings of our study hold significant implications for both precision agriculture and potato-breeding programs. In the context of precision agriculture, the ability to accurately monitor water availability provides a valuable tool for optimizing resource allocation and irrigation management, using variable-rate application. This ensures that water is supplied precisely where and when it is needed, maximizing yield potential. Moreover, our results offer a non-destructive means of assessing crop water condition, enabling mitigation of potential losses. For potato-breeding programs, our findings offer a potential tool for accelerating the selection of drought-tolerant cultivars. By leveraging hyperspectral imaging, breeders can rapidly screen and identify promising varieties that exhibit robust responses to varying water conditions. This expedites the breeding process, allowing for the development of drought-resistant potatoes that are well suited to changing climatic conditions. The ability to phenotype crops more comprehensively and efficiently enhances the precision and success of breeding efforts, ultimately contributing to the creation of improved cultivars that ensure food security and sustainable agricultural practices. The acquired data could be used for further development of remote sensing applications enabling targeted water management and plant health assessment. These could prevent the development and spread of soil-borne pests [21,73] that are emerging due to climate change causing higher average temperatures.

Even though the results of this study are promising, several noteworthy factors should be considered. The study’s execution within a controlled laboratory environment introduces a potential constraint, as it becomes challenging to extrapolate the model’s performance accurately onto real-world field conditions. Therefore, the research should be repeated in the field to evaluate the impact on the performance. An additional limitation involves the ground truth labeling for both varieties. The reliance on implicit exposure through soil moisture measurements, facilitated by tensiometers, introduces ambiguity regarding the actual water-deficit conditions experienced by the underlying potato plants. An alternative, more precise approach, could involve the direct measurement of water potential, despite its inherently destructive nature.

The size of the self-acquired dataset also poses a constraint due to factors such as cost, spatial limitations, and time constraints. The relatively small dataset utilized in this study, while necessary due to these practical considerations, leaves room for a larger dataset to be collected in order to achieve more precise estimates and confidence intervals [74]. This expansion would contribute to solidifying the robustness of the findings. Moreover, the preprocessing step involved the rescaling of SWIR images to guarantee spectral channel concatenation compatibility. The process of rescaling, though necessary, introduces the possibility of minor misalignments between the images, potentially leading to suboptimal model performance. Hence, employing even more robust techniques in the next iteration could further enhance the data preprocessing pipeline. While the study provides valuable insights, its limitations emphasize the necessity for careful result interpretation and indicate potential opportunities for further research to enhance the study’s applicability.

## 5. Conclusions

In this study, we showed how a combination of attention-based deep neural network and hyperspectral imaging can be used for the detection of water-deficit conditions by applying it to two potato plant varieties with different drought-tolerance capabilities. Although well-watered and water-deficient potato plants after the experiment looked the same to the naked eye, we still managed to achieve sufficient classification results. Moreover, we demonstrated how different sampling strategies and the introduction of biases influence the classification metrics, showing that the exploitation of deep learning models requires careful adoption to be effectively employed on hyperspectral datasets. In addition to this, we investigated which spectral channels of input data provided the most information for the successful modeling of the presence of water deficiency. Given the study’s reliance on a constrained dataset, additional data and analysis are needed to fully confirm the findings. Additionally, by reproducing the research in real-world settings, the study’s applicability could be expanded in the future.

## Figures and Tables

**Figure 1 plants-13-01918-f001:**
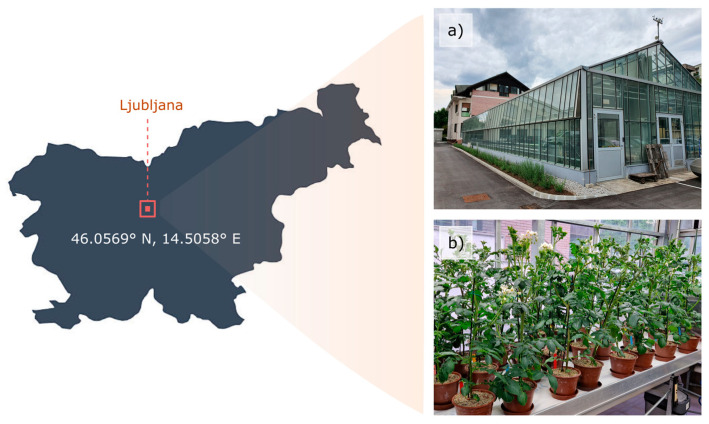
Location of the experiment: (**a**) The greenhouse where the experiment was conducted and (**b**) potato plants in a growth chamber.

**Figure 2 plants-13-01918-f002:**
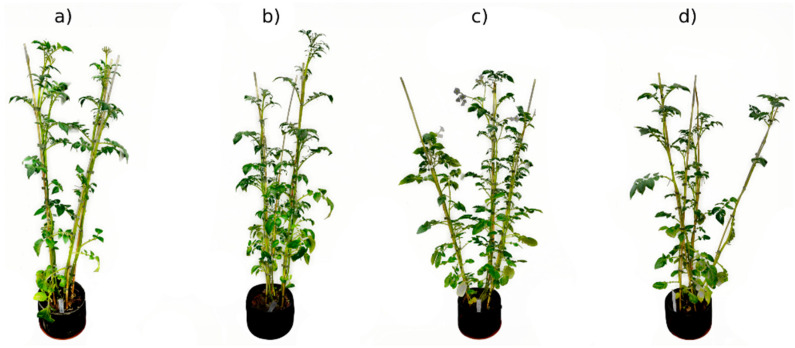
Plants of both cultivars at the end of experiment. (**a**) Well-watered KIS Krka, (**b**) water-deficient KIS Krka, (**c**) well-watered KIS Savinja, and (**d**) water-deficient KIS Savinja. Plants from neither cultivar showed exceedingly expressed symptoms of drought stress.

**Figure 3 plants-13-01918-f003:**
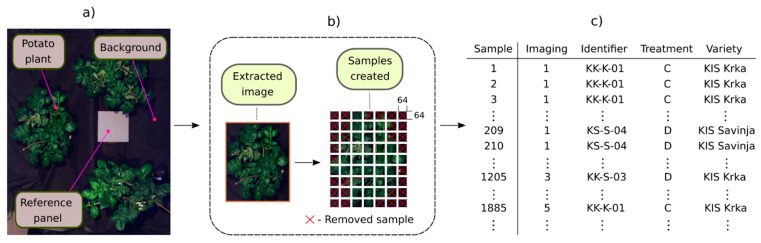
(**a**) Original unsegmented hyperspectral image; (**b**) SiaPy transformation process, where each image is segmented, potato plant extracted, and sliced to multiple samples. RGB (red–green–blue) color scheme is represented using VNIR 55th, 41st, and 12th (610 nm, 559 nm, and 453 nm) spectral bands; (**c**) segments of sample attributes.

**Figure 4 plants-13-01918-f004:**
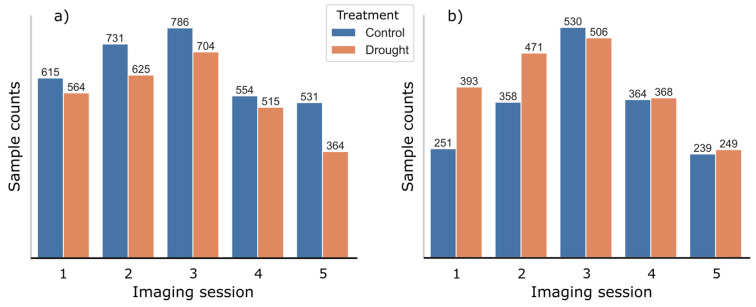
Number of well-watered (control) and water-deficient (drought) samples per imaging session for (**a**) KIS Krka and (**b**) KIS Savinja.

**Figure 5 plants-13-01918-f005:**
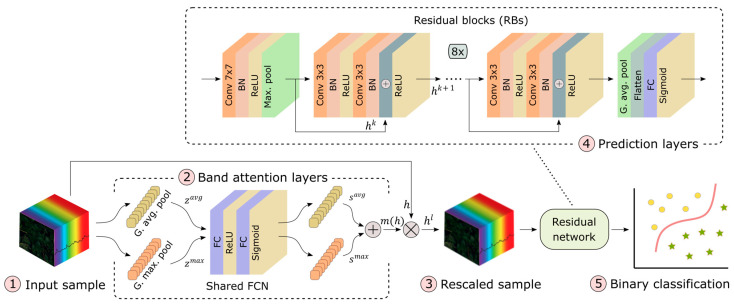
Architecture of end-to-end deep learning model. Conv—convolution; BN—batch normalization; ReLU—rectified linear unit; G. max. pool—global maximum pooling; G. avg. pool—global average pooling; FC—fully connected.

**Figure 6 plants-13-01918-f006:**
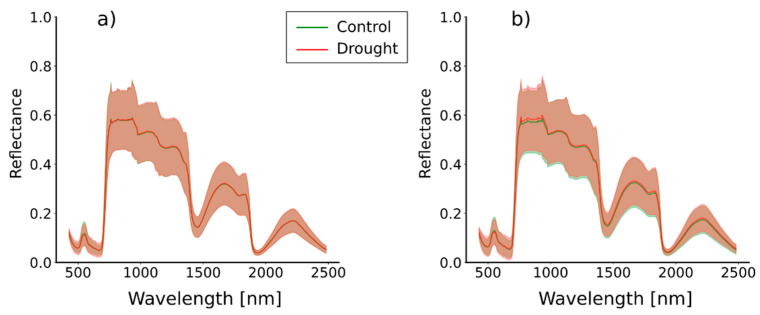
Mean spectral signatures for (**a**) KIS Krka and (**b**) KIS Savinja. Mean value and standard deviation are represented with solid line and ribbon, respectively.

**Figure 7 plants-13-01918-f007:**
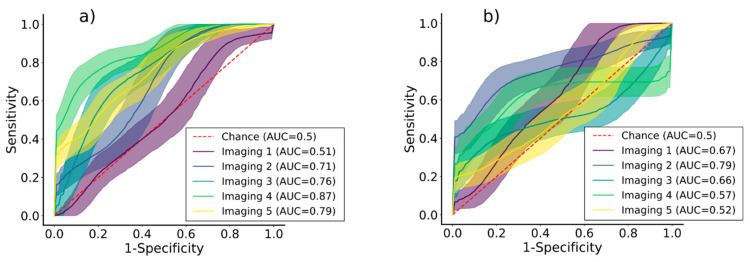
ROC curves for varieties (**a**) KIS Krka and (**b**) KIS Savinja, generated based on UD-SS. Mean values and standard deviations are represented with solid lines and ribbons, respectively. Each imaging session is depicted with one arbitrary color and random prediction with dashed red line. Mean values of AUCs are written inside brackets next to annotations of imaging sessions.

**Figure 8 plants-13-01918-f008:**
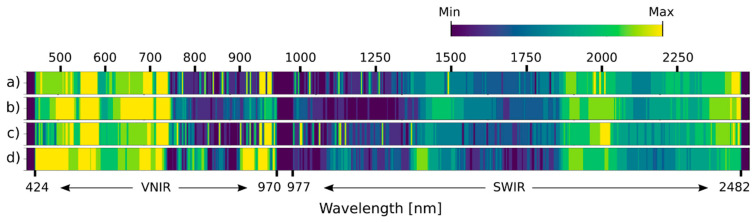
Relevance of spectral bands as determined by band attention module for KIS Krka. (**a**) UD-SS, (**b**) UD-RS, (**c**) 3. BDT-SS, and (**d**) BDI-SS. The brighter the color, the more significant the spectral channel.

**Figure 9 plants-13-01918-f009:**
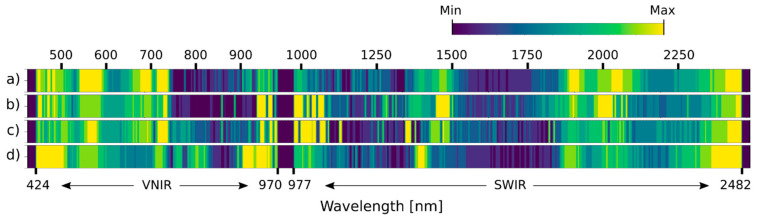
Relevance of spectral bands as determined by band attention module, for KIS Savinja. (**a**) UD-SS, (**b**) UD-RS, (**c**) 3. BDT-SS, and (**d**) BDI-SS. The brighter the color, the more significant the spectral channel.

**Figure 10 plants-13-01918-f010:**
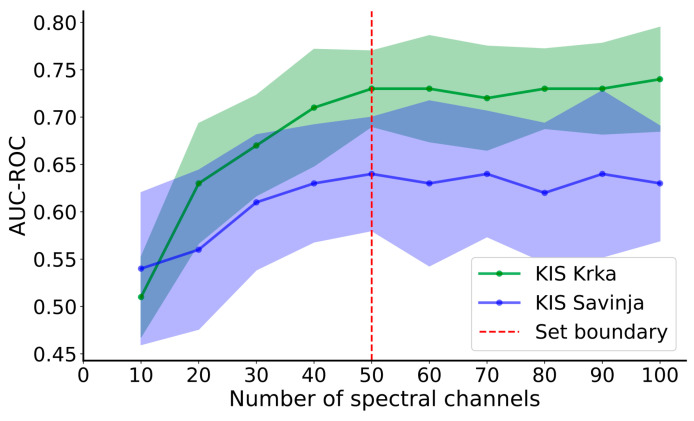
AUC-ROC metric plotted against the count of selected spectral channels. The data for KIS Krka are depicted in green and for KIS Savinja in blue. The dashed red line signifies the set boundary. Mean values and standard deviations are represented with solid lines and ribbons, respectively.

**Table 1 plants-13-01918-t001:** Classification results for KIS Krka based on UD-SS. The performance was assessed using various metrics with 95% confidence intervals written in brackets. First two columns denote the number of unique samples (#) per treatment (C or D) that were used for training (train) and testing (test).

Imaging	# C/D—Train	# C/D—Test	AUC-ROC	F1 Score	Precision	Recall
1	306/306	58/58	0.51 (0.40, 0.62)	0.54 (0.45, 0.63)	0.54 (0.45, 0.63)	0.54 (0.44, 0.63)
2	306/306	58/58	0.71 (0.59, 0.79)	0.68 (0.59, 0.76)	0.71 (0.62, 0.78)	0.69 (0.59, 0.76)
3	306/306	58/58	0.76 (0.66, 0.84)	0.65 (0.56, 0.74)	0.69 (0.59, 0.76)	0.66 (0.58, 0.74)
4	306/306	58/58	0.87 (0.80, 0.92)	0.78 (0.69, 0.84)	0.78 (0.69, 0.84)	0.78 (0.68, 0.84)
5	306/306	58/58	0.79 (0.70, 0.86)	0.70 (0.60, 0.77)	0.70 (0.60, 0.77)	0.70 (0.59, 0.77)
Pooled	1530/1530	290/290	0.74 (0.70, 0.78)	0.67 (0.64, 0.71)	0.68 (0.64, 0.71)	0.67 (0.64, 0.71)

**Table 2 plants-13-01918-t002:** Classification results for KIS Savinja based on UD-SS. The performance was assessed using various metrics with 95% confidence intervals written in brackets. First two columns denote the number of unique samples (#) per treatment (C or D) that were used for training (train) and testing (test).

Imaging	# C/D—Train	# C/D—Test	AUC-ROC	F1 Score	Precision	Recall
1	151/151	30/30	0.67 (0.51, 0.78)	0.62 (0.47, 0.72)	0.62 (0.48, 0.72)	0.62 (0.47, 0.72)
2	151/151	30/30	0.79 (0.65, 0.89)	0.76 (0.63, 0.86)	0.80 (0.67, 0.87)	0.77 (0.63, 0.85)
3	151/151	30/30	0.66 (0.51, 0.80)	0.65 (0.53, 0.77)	0.65 (0.51, 0.76)	0.65 (0.52, 0.75)
4	151/151	30/30	0.57 (0.40, 0.72)	0.60 (0.44, 0.72)	0.71 (0.54, 0.80)	0.63 (0.50, 0.73)
5	151/151	30/30	0.52 (0.36, 0.66)	0.49 (0.35, 0.62)	0.50 (0.33, 0.62)	0.50 (0.35, 0.60)
Pooled	755/755	150/150	0.64 (0.56, 0.69)	0.63 (0.56, 0.68)	0.64 (0.58, 0.69)	0.63 (0.57, 0.68)

**Table 3 plants-13-01918-t003:** Classification results for majority voting for both varieties. based on UD-SS. The performance was assessed using various metrics with 95% confidence intervals written in brackets. First two columns denote the number of unique samples (#) per treatment (C or D) that were used for training (train) and testing (test).

Variety	# C/D—Train	# C/D—Test	AUC-ROC	F1 Score	Sensitivity	Specificity
KIS Krka	1530/1530	290/290	0.89 (0.69, 0.97)	0.73 (0.52, 0.87)	0.74 (0.50, 0.86)	0.73 (0.50, 0.83)
KIS Savinja	755/755	150/150	0.56 (0.24, 0.83)	0.73 (0.44, 0.90)	0.83 (0.76, 0.91)	0.75 (0.45, 0.85)

**Table 4 plants-13-01918-t004:** Classification results for KIS Krka based on UD-RS. The performance was assessed using various metrics with 95% confidence intervals written in brackets. First two columns denote the number of unique samples (#) per treatment (C or D) that were used for training (train) and testing (test).

Imaging	# C/D—Train	# C/D—Test	AUC-ROC	F1 Score	Precision	Recall
1	276/276	54/54	0.58 (0.47, 0.68)	0.51 (0.42, 0.60)	0.51 (0.40, 0.59)	0.51 (0.40, 0.59)
2	276/276	54/54	0.65 (0.55, 0.76)	0.57 (0.48, 0.67)	0.57 (0.48, 0.67)	0.57 (0.47, 0.66)
3	276/276	54/54	0.85 (0.76, 0.91)	0.76 (0.68, 0.83)	0.76 (0.67, 0.83)	0.76 (0.67, 0.82)
4	276/276	54/54	0.77 (0.68, 0.85)	0.65 (0.55, 0.74)	0.65 (0.55, 0.74)	0.65 (0.55, 0.73)
5	276/276	54/54	0.81 (0.73, 0.88)	0.75 (0.67, 0.82)	0.75 (0.67, 0.82)	0.75 (0.67, 0.81)
Pooled	1380/1380	270/270	0.74 (0.70, 0.78)	0.65 (0.61, 0.69)	0.65 (0.61, 0.69)	0.65 (0.61, 0.69)

**Table 5 plants-13-01918-t005:** Classification results for KIS Savinja based on UD-RS. The performance was assessed using various metrics with 95% confidence intervals written in brackets. First two columns denote the number of unique samples (#) per treatment (C or D) that were used for training (train) and testing (test).

Imaging	# C/D—Train	# C/D—Test	AUC-ROC	F1 Score	Precision	Recall
1	191/191	38/38	0.87 (0.76, 0.93)	0.73 (0.61, 0.82)	0.77 (0.64, 0.85)	0.74 (0.62, 0.82)
2	191/191	38/38	0.90 (0.81, 0.96)	0.83 (0.72, 0.91)	0.83 (0.73, 0.91)	0.83 (0.71, 0.89)
3	191/191	38/38	0.83 (0.73, 0.90)	0.70 (0.58, 0.79)	0.70 (0.58, 0.79)	0.70 (0.57, 0.78)
4	191/191	38/38	0.88 (0.79, 0.94)	0.79 (0.68, 0.87)	0.79 (0.67, 0.86)	0.79 (0.67, 0.86)
5	191/191	38/38	0.91 (0.81, 0.96)	0.80 (0.70, 0.88)	0.80 (0.69, 0.87)	0.80 (0.68, 0.87)
Pooled	955/955	190/190	0.87 (0.84, 0.91)	0.77 (0.73, 0.81)	0.77 (0.73, 0.81)	0.77 (0.72, 0.81)

**Table 6 plants-13-01918-t006:** Classification results for KIS Krka based on BDT-SS. The performance was assessed using various metrics with 95% confidence intervals written in brackets. First two columns denote the number of unique samples (#) per treatment (C or D) that were used for training (train) and testing (test).

Imaging	# C/D—Train	# C/D—Test	AUC-ROC	F1 Score	Precision	Recall
1	306/61	58/11	0.41 (0.25, 0.56)	0.75 (0.62, 0.85)	0.70 (0.54, 0.83)	0.80 (0.68, 0.87)
2	306/61	58/11	0.68 (0.53, 0.80)	0.77 (0.61, 0.85)	0.71 (0.53, 0.81)	0.84 (0.72, 0.90)
3	183/183	34/34	0.88 (0.77, 0.95)	0.81 (0.69, 0.88)	0.82 (0.70, 0.88)	0.81 (0.68, 0.88)
4	61/306	11/58	0.71 (0.50, 0.86)	0.84 (0.71, 0.91)	0.84 (0.69, 0.91)	0.86 (0.74, 0.91)
5	61/306	11/58	0.71 (0.48, 0.86)	0.82 (0.70, 0.91)	0.81 (0.67, 0.90)	0.84 (0.72, 0.90)
Pooled	917/917	172/172	0.87 (0.83, 0.90)	0.83 (0.78, 0.86)	0.83 (0.79, 0.86)	0.83 (0.78, 0.86)

**Table 7 plants-13-01918-t007:** Classification results for KIS Savinja based on BDT-SS. The performance was assessed using various metrics with 95% confidence intervals written in brackets. First two columns denote the number of unique samples (#) per treatment (C or D) that were used for training (train) and testing (test).

Imaging	# C/D—Train	# C/D—Test	AUC-ROC	F1 Score	Precision	Recall
1	30/151	6/30	0.59 (0.38, 0.71)	0.71 (0.51, 0.85)	0.68 (0.43, 0.84)	0.75 (0.56, 0.83)
2	30/151	6/30	0.73 (0.59, 0.82)	0.76 (0.50, 0.88)	0.69 (0.41, 0.84)	0.83 (0.64, 0.92)
3	90/90	18/18	0.41 (0.23, 0.63)	0.44 (0.29, 0.61)	0.44 (0.27, 0.60)	0.44 (0.28, 0.58)
4	151/30	30/6	0.51 (0.05, 0.95)	0.82 (0.62, 0.92)	0.81 (0.56, 0.92)	0.83 (0.64, 0.92)
5	151/30	30/6	0.50 (0.32, 0.75)	0.70 (0.52, 0.83)	0.71 (0.49, 0.87)	0.69 (0.50, 0.81)
Pooled	452/452	90/90	0.76 (0.68, 0.83)	0.71 (0.64, 0.77)	0.71 (0.64, 0.78)	0.71 (0.64, 0.77)

**Table 8 plants-13-01918-t008:** Classification results for KIS Krka based on BDI-SS. The performance was assessed using various metrics with 95% confidence intervals written in brackets. First two columns denote the number of unique samples (#) per treatment (C or D) that were used for training (train) and testing (test).

Imaging	# C/D—Train	# C/D—Test	AUC-ROC	F1 Score	Precision	Recall
1	61/61	11/11	0.34 (0.12, 0.62)	0.34 (0.15, 0.58)	0.34 (0.11, 0.60)	0.36 (0.14, 0.55)
2	61/61	11/11	0.95 (0.75, 1.00)	0.81 (0.60, 0.95)	0.87 (0.80, 0.96)	0.82 (0.59, 0.91)
3	183/183	34/34	0.85 (0.75, 0.93)	0.79 (0.69, 0.87)	0.80 (0.69, 0.87)	0.79 (0.67, 0.87)
4	306/306	58/58	0.86 (0.78, 0.92)	0.78 (0.69, 0.84)	0.78 (0.69, 0.85)	0.78 (0.68, 0.84)
5	306/306	58/58	0.80 (0.71, 0.88)	0.74 (0.66, 0.82)	0.77 (0.69, 0.84)	0.75 (0.66, 0.81)
Pooled	917/917	172/172	0.82 (0.77, 0.86)	0.75 (0.70, 0.79)	0.75 (0.70, 0.79)	0.75 (0.70, 0.79)

**Table 9 plants-13-01918-t009:** Classification results for KIS Savinja based on BDI-SS. The performance was assessed using various metrics with 95% confidence intervals written in brackets. First two columns denote the number of unique samples (#) per treatment (C or D) that were used for training (train) and testing (test).

Imaging	# C/D—Train	# C/D—Test	AUC-ROC	F1 Score	Precision	Recall
1	151/151	30/30	0.82 (0.70, 0.91)	0.71 (0.60, 0.82)	0.72 (0.60, 0.82)	0.72 (0.60, 0.80)
2	151/151	30/30	0.72 (0.57, 0.83)	0.65 (0.52, 0.77)	0.65 (0.51, 0.77)	0.65 (0.50, 0.75)
3	90/90	18/18	0.59 (0.38, 0.77)	0.52 (0.34, 0.67)	0.53 (0.33, 0.67)	0.53 (0.33, 0.64)
4	30/30	6/6	0.28 (0.01, 0.69)	0.29 (0.07, 0.60)	0.23 (0.04, 0.55)	0.42 (0.08, 0.67)
5	30/30	6/6	0.50 (0.09, 0.86)	0.58 (0.24, 0.83)	0.59 (0.17, 0.83)	0.58 (0.17, 0.75)
Pooled	452/452	90/90	0.67 (0.57, 0.74)	0.63 (0.54, 0.69)	0.63 (0.54, 0.69)	0.63 (0.54, 0.69)

**Table 10 plants-13-01918-t010:** Classification results for KIS Krka based on UD-SS with reduced spectral bands. The performance was assessed using various metrics with 95% confidence intervals written in brackets. First two columns denote the number of unique samples (#) per treatment (C or D) that were used for training (train) and testing (test).

Imaging	# C/D—Train	# C/D—Test	AUC-ROC	F1 Score	Precision	Recall
1	306/306	58/58	0.29 (0.21, 0.40)	0.34 (0.25, 0.44)	0.34 (0.23, 0.45)	0.41 (0.30, 0.49)
2	306/306	58/58	0.76 (0.67, 0.84)	0.67 (0.58, 0.75)	0.67 (0.58, 0.75)	0.67 (0.58, 0.74)
3	306/306	58/58	0.78 (0.69, 0.85)	0.68 (0.58, 0.76)	0.69 (0.59, 0.76)	0.68 (0.58, 0.75)
4	306/306	58/58	0.84 (0.76, 0.90)	0.78 (0.69, 0.84)	0.78 (0.70, 0.85)	0.78 (0.68, 0.84)
5	306/306	58/58	0.83 (0.75, 0.89)	0.71 (0.62, 0.78)	0.72 (0.63, 0.79)	0.72 (0.62, 0.78)
Pooled	1530/1530	290/290	0.73 (0.68, 0.76)	0.65 (0.61, 0.69)	0.65 (0.61, 0.69)	0.65 (0.61, 0.69)

**Table 11 plants-13-01918-t011:** Classification results for KIS Savinja based on UD-SS with reduced spectral bands. The performance was assessed using various metrics with 95% confidence intervals written in brackets. First two columns denote the number of unique samples (#) per treatment (C or D) that were used for training (train) and testing (test).

Imaging	# C/D—Train	# C/D—Test	AUC-ROC	F1 Score	Precision	Recall
1	151/151	30/30	0.78 (0.65, 0.88)	0.72 (0.60, 0.82)	0.72 (0.58, 0.82)	0.72 (0.58, 0.80)
2	151/151	30/30	0.74 (0.58, 0.85)	0.62 (0.46, 0.73)	0.65 (0.48, 0.75)	0.63 (0.47, 0.72)
3	151/151	30/30	0.55 (0.40, 0.70)	0.50 (0.36, 0.62)	0.50 (0.35, 0.61)	0.50 (0.35, 0.60)
4	151/151	30/30	0.47 (0.32, 0.63)	0.60 (0.45, 0.72)	0.71 (0.51, 0.79)	0.63 (0.48, 0.73)
5	151/151	30/30	0.72 (0.55, 0.84)	0.59 (0.45, 0.71)	0.61 (0.47, 0.72)	0.60 (0.45, 0.70)
Pooled	755/755	150/150	0.64 (0.58, 0.70)	0.61 (0.54, 0.66)	0.63 (0.57, 0.68)	0.62 (0.55, 0.66)

## Data Availability

In line with our commitment to fostering open-science practices, we have made the data and code readily accessible. Interested researchers will be able to access the pre-processed dataset at https://doi.org/10.5281/zenodo.7936850 (accessed on 8 July 2024) and code at https://github.com/janezlapajne/manuscripts (accessed on 8 July 2024). By providing transparent and unrestricted access to these resources, we aim to encourage collaborative replication, validation, and further exploration of the findings presented in this paper. The accompanying documentation ensures that scientists can seamlessly engage with the materials, promoting transparency and reproducibility.

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
