# Peer review of "Enhancing Water-Deficient Potato Plant Identification: Assessing Realistic Performance of Attention-Based Deep Neural Networks and Hyperspectral Imaging for Agricultural Applications"

_plants, 2024, doi:10.3390/plants13141918_

Round 1
Reviewer 1 Report
Comments and Suggestions for Authors
The objective of this research was to assess the performance of deep learning models on evaluating the effect of water deficit on two potato cultivars. It is proposed a new Hyperspectral imaging (HSI)-based method using attention-based deep learning for detection of water-deficiency in potato plants. Data from two potato varieties were used to evaluate the classification performance over several weeks, and to test capability for early detection. HSI is a non-invasive technique that captures an object's spectral and spatial features through imaging at various wavelengths. The optimal spectral ranges for analyzing plants and vegetation are visible and near-infrared ranges, which enables the assessment of variations in leaf pigmentation (400-700 nm) and mesophyll cell structure (700-1300 nm), while longer wavelengths (1300-2500 nm) are required to detect changes in plant water content. It was found that the evaluation of performance on datasets with deliberately introduced biases consistently demonstrated superior results in comparison to their non-biased equivalents. Also, it is reported that the wavelengths of greatest significance were observed in the ranges of 475–580 nm, 660–730 nm, 940–970 nm, 1420–1510 nm, 1875–2040 nm, and 2350–2480 nm.
One of the contributions of this paper is to show that both VNIR and SWIR parts of the spectrum are relevant to effectively distinguish between well-watered and water-deficient potato plants.
The manuscript is well-written and can be of interest to the readership of Plants. The reviewer suggests a few amendments which aim to improve the quality of the manuscript:
Materials and Methods
= Tensiometers were used to monitor the soil moisture levels. Throughout the duration of the experiment, the matric potential of the soil was maintained within the range of -0.01 MPa to -0.025 MPa for well-watered plants and from -0.05 MPa to -0.07 MPa for water-deficient plants =
If available, the reviewer suggests adding the leaf water potential (pre-dawn and midday leaf water potential), as leaf water potential may differ from soil water potential depending on vapor pressure deficit (i.e. atmosphere dryness) .
= Figure 3. Distribution of well-watered (control) and water-deficient (drought) =
Suggestion: if data are available: it is suggested to add the standard error to the bars.
Others:
It is suggested to add a list of abbreviations used in the manuscript; e.g.: AUC-ROC: area under the … curve; SWIR: ..., VNIR: … , and so on.
Author Response
Comments 1:
Tensiometers were used to monitor the soil moisture levels. Throughout the duration of the experiment, the matric potential of the soil was maintained within the range of -0.01 MPa to -0.025 MPa for well-watered plants and from -0.05 MPa to -0.07 MPa for water-deficient plants = If available, the reviewer suggests adding the leaf water potential (pre-dawn and midday leaf water potential), as leaf water potential may differ from soil water potential depending on vapor pressure deficit (i.e. atmosphere dryness).
Response 1:
Thank you for the suggestion. Another reviewer also highlighted a similar point, prompting me to include the midday leaf water potential and its measured values. However, due to constraints in cost and time, we did not measure the pre-dawn potential. You can find the refined text in the Plants Growing Setup section and the measurements in the Appendix section.
Comments 2:
= Figure 3. Distribution of well-watered (control) and water-deficient (drought) Suggestion: if data are available: it is suggested to add the standard error to the bars.
Response 2:
The error bars are not needed in this case since the number is exact—it represents the total number of available data instances, categorized by imaging session and treatment condition. I understand why you suggested adding error bars; I mistakenly labeled it as "Distribution" instead of "Number." I have now corrected this.
Comments 3:
It is suggested to add a list of abbreviations used in the manuscript; e.g.: AUC-ROC: area under the … curve; SWIR: ..., VNIR: … , and so on.
Response 3:
Each abbreviation is written in its long form at its first mention, as per the usual practice in this journal. However, if the editor requests a separate table of abbreviations, I am prepared to create one. Below are the author guidelines for this journal:
Reviewer 2 Report
Comments and Suggestions for Authors
This article evaluates the application performance of deep learning models on two potato varieties exposed to water scarcity conditions. By using a dual sensor hyperspectral imaging system, we explored how various sampling strategies and biases affect classification metrics. In addition, it will focus on accurately locating the key wavelengths that indicate water scarcity in the connected images. The proposed deep learning model has produced very good results. I suggest publishing this article after some revisions.
1, The author should clearly state in the materials and methods the irrigation conditions under drought stress and normal control. What are the drought conditions used in this article?
2, Why was only one pot of plants placed in Figure 1? It's best to put three repetitions together for a photo.
3, Academic paper data should be subjected to significance analysis, shouldn't data in Figure 3 be subjected to significance analysis?
Author Response
Comments 1:
The author should clearly state in the materials and methods the irrigation conditions under drought stress and normal control. What are the drought conditions used in this article?
Response 1:
Thank you for expressing your concern about how we determined the water-deficient conditions, as this is indeed not a trivial task. The primary method for inducing stress in the plants was through direct measurement of matric potential using tensiometers, as outlined in the initial draft, since this method allowed us to quickly define the amount of water provided to the plants. The ranges defined in the paper were established through consultation with professional personnel and thorough literature review. For example, similar stress levels are reported in this study: https://doi.org/10.1007/s12230-012-9291-y, where they state:
… SMP of −25 kPa was the most favorable setting for potato production, while −15 kPa was too high and −45 kPa lead to severe water stress …
Therefore, the defined values guided us to ensure certain plant conditions. Additionally, the environmental variables were consistently maintained using our glasshouse system, which kept a relative humidity of 60% and temperatures of 15°C (night) and 21°C (day). This ensured that our experimental conditions were stable and reproducible, allowing for accurate assessment of plant responses under controlled stress conditions. To emphasize stable environmental conditions, I have included the tolerances for temperature and humidity levels in the revised manuscript.
Furthermore, as an additional check, leaf water potential was measured in potato plants concurrently with the imaging sessions. Initially, these measurements were conducted as a "sanity" check and were not included in the paper. However, in the revised manuscript, I have expanded the Plants Growing Setup section to include this information. Additionally, I have added the measured values and corresponding statistics to the Appendix. The calculated p-values indicate a significant difference between the two treatments across all imaging sessions. To further clarify, I have placed this information in the Appendix because it is not the main focus of the paper and has been already researched in other studies. However, I agree that including it provides readers with a specific reference for what constitutes control and drought conditions in our study.
I hope this addresses the concern raised. If further clarification is needed, I can provide additional information, justification, and reasoning behind the execution of the experiment.
Comments 2:
Why was only one pot of plants placed in Figure 1? It's best to put three repetitions together for a photo.
Response 2:
The main purpose of Figure 1, which displays both well-watered and water-deficient potato varieties, is to demonstrate that the plants appeared visually similar to the human eye. Our goal was to provide the reader with a clear visual representation of the plants for a better overall understanding. Unfortunately, we do not have a side image of the potato plants that shows all three repetitions. We hopefully believe that the top-down view of the repetitions, as presented in Figure 2, is sufficient.
Comments 3:
Academic paper data should be subjected to significance analysis, shouldn't data in Figure 3 be subjected to significance analysis?
Response 3:
I agree that significance analysis should be included in any research where applicable, especially when direct measurements of each sample are available (e.g., measurements of physiological values, LAI, SPAD, etc.). However, in this study, the plant images were further divided into smaller samples to generate a sufficient dataset for training the deep neural network (note: samples from the same plant were not used for both training and testing). As direct measurements for each individual sample are not available, it is not possible to perform significance analysis at the sample level. In response to Comment 1, I have included significance analysis between the two treatments at the plant level, but a similar analysis cannot be conducted at the sample level.
However, to objectively measure the difference between the two treatments, we applied a methodology commonly used in the machine learning field—evaluating metrics, primarily the F1 score, to assess how accurately the model distinguishes between the treatments. To provide a comprehensive evaluation beyond just the mean metric values, we calculated 95% confidence intervals (CI) for all metrics using bootstrapping with a thousand resamples, similar to methodology employed in https://www.nature.com/articles/s41746-021-00553-x .
Additionally, for readers interested in further exploring the data, we have provided the code with examples. The link has already been provided, but for quick reference, here is the link to the exploration scripts/notebooks: https://github.com/Manuscripts-code/Potato-plants-drought--MDPI-2024/tree/main/notebooks

Reviewer 3 Report
Comments and Suggestions for Authors
This study proposes a strategy of integration of attention-based deep learning models for the classification by using a dual-sensor hyperspectral imaging system. The model performance is validated on two potato plant cultivars exposed to water-deficient condition. It explores how various sampling strategies and biases impact. This work is practical, and has potential to benefit readers. I recommend authors make revisions to improve the manuscript.
Please give a map to show the locations where the samples are collected.
Hyperspectral analysis is sensitive to ROI, please clarify the clever manner you used to extract the data from the rough hyperspectral map.
Hyperspectral imaging is prevalent to use the VNIR and SWIR bands. The model is much related to the chemometric methods. Please highlight the novelty of your study on algorithms.
Please give details about the principle of band attention mechanism.
Please share the code for modeling.
The conclusion section should be rewritten, in combined discussion with your results.
There are too many reference articles. A research paper conventionally uses around 40 reference articles.
Comments on the Quality of English LanguageModerate editing of English language required
Author Response
Comments 1:
Please give a map to show the locations where the samples are collected.
Response 1:
The map was added to the revised manuscript.
Comments 2:
Hyperspectral analysis is sensitive to ROI, please clarify the clever manner you used to extract the data from the rough hyperspectral map.
Response 2:
Thank you very much for the comment. We unintentionally forgot to add the citation of our previously published paper (https://doi.org/10.3390/s22010367), which thoroughly explains the methodology (see chapter 2.3. Pre-Processing and Analysis). Furthermore, I have also included a conference paper which might be beneficial to the reader (https://www.wageningenacademic.com/doi/10.3920/978-90-8686-947-3_54) on the SiaPy software used in our study, where the procedure is implemented. Both references have been added to the revised paper.
Comments 3:
Hyperspectral imaging is prevalent to use the VNIR and SWIR bands. The model is much related to chemometric methods. Please highlight the novelty of your study on algorithms.
Response 3:
As you noted, our method shares similarities with conventional chemometric approaches that typically make predictions based on 1D spectra. However, our study applies the model directly to 2D spectra, while still enabling the extraction of relevant spectral areas through the use of a band-attention module. This approach combines the enhanced modeling capabilities over conventional chemometric methods, but still enabling the convenience of directly extracting relevant spectral features.
The core concepts are already outlined in the paper in a more generalized form, without specifically focusing on chemometric methods. For instance, the introduction discusses the advantages of deep neural networks compared to standard methods such as LDA, PLS, and PCA (which may be also used for chemometric purposes):
… The emergence of deep learning has revolutionized the field of machine learning, proving to be highly effective in solving complex problems. Deep learning has outperformed traditional models in domains such as image and speech recognition, and natural language processing [27]. Its superior performance has also been demonstrated in hyper-spectral image (HSI) classification tasks [28,29]. One essential architecture in deep learning is the Residual Neural Network (ResNet), which employs skip connections or shortcuts to overcome vanishing gradient problems, making neural network optimization easier [30] …
The novelty of using the proposed algorithm over conventional methods, including classic chemometric methods, is discussed in the paper's Discussion section:
… Our results demonstrate that it is possible to combine hyperspectral imaging data with the proposed attention-based deep neural network to successfully discriminate be-tween water-deficient potato plants. The challenge in distinguishing between both treatments for either variety is underscored by the similarity of their average spectral signatures (Figure 5). This highlights the potential difficulty of solely relying on spectral information for differentiation, as done in [22]. However, leveraging the power of deep neural networks, which exploit combined spectral-spatial information, allows for the discernment of differences based on the spatial distribution of spectral signatures within a sample [38]. Therefore, it can provide valuable information about the plant's physiological and bio-chemical characteristics …
I hope I have clarified the significant advantages of using deep neural networks over traditional methods for analyzing spectral data and addressed the question posed.
Comments 4:
Please give details about the principle of band attention mechanism.
Response 4:
I'm unsure which aspect of the band attention mechanism still requires clarification. The mathematical principles of its operation are detailed in Chapter 2.7, "Band Attention Mechanism" (I added "mechanism" to reduce ambiguity). The procedure for extracting the most relevant spectral bands is explained in Chapter 2.9, "Procedure for Selection of Prominent Spectral Bands." The confusion may came from the order in which the method is presented. I structured the description based on Figure 4, where I outlined the neural network components from left to right. If this explanation does not address your question, please let me know specifically which part requires further clarification.
Comments 5:
Please share the code for modeling.
Response 5:
This is the link to the paper’s code: https://github.com/Manuscripts-code/Potato-plants-drought--MDPI-2024 . The link is already provided and accessible in the article text. Regarding the data, it will be published upon acceptance of the article. At that time, the missing parts of the code README file will also be completed.
Comments 6:
The conclusion section should be rewritten, in combined discussion with your results.
Response 6:
I'm uncertain whether this is a suggestion or a mandatory requirement. I followed the general layout by providing a summary of the results and discussion, and suggesting future research directions. I also checked the journal's guidelines, which state that a conclusion section is not mandatory. The guidelines specify: "This section is not mandatory but can be added to the manuscript if the discussion is unusually long or complex." Therefore, I hope the conclusion can remain as it is. If you disagree, please provide more specific feedback or an example of how I should revise the conclusion or integrate new information.
Comments 7:
There are too many reference articles. A research paper conventionally uses around 40 reference articles.
Response 7:
Upon reviewing the author’s guidelines, I did not find any constraints on the number of references. Additionally, when searching through the Web of Science, I found many research articles that exceed our current number of references. For example, you can refer to https://doi.org/10.3390/plants11111425, which includes over 100 references. Therefore, I hope that reducing the number of references in our case is not mandatory, especially considering that all the literature was carefully selected to support our statements.

Round 2
Reviewer 2 Report
Comments and Suggestions for Authors
Authors have addressed all my comments.